# LVM-Net: Efficient Long-Form Video Reasoning using Neural Sampling

## Abstract

Long-form video reasoning is essential for various applications such as video retrieval, summarizing, and question answering. However, existing methods often require significant computational resources and are limited by GPU memory constraints. To address this challenge, we present Long-Video Memory Network, *LVM-Net*, a novel video reasoning method that employs a fixed-size memory representation to store discriminative patches sampled from the input video. By leveraging a *neural sampler* that identifies discriminative memory tokens, LVM-Net achieves improved efficiency. Furthermore, LVM-Net only requires a single pass over the video, further enhancing overall efficiency. Our results on the Rest-ADL dataset demonstrate an 18x – 75x improvement in inference times for long-form video retrieval and answering questions, with a competitive predictive performance.

## 1 Introduction

Long-form video understanding is important for various applications such as video retrieval, summarizing, and question answering. For example, for automated checkout in retail applications, a video understanding system needs to understand the temporal order of important actions such as grabbing an object, and limit the attention to actions such as browsing items, and interacting with other people, to efficiently process long duration shopping videos (Wankhede et al., 2018; Strafforello et al., 2023).

Modern methods for long-form video understanding such as transformer based models can be inefficient and require significant computational resources (Wu & Krahenbuhl, 2021). These methods often require building an intermediate representation for the entire video in memory and consume large amounts of GPU memory limiting the maximum length of videos that can be processed, especially for reasoning tasks that require joint spatio-temporal analysis of different scene elements (Fournier et al., 2023). These tasks necessitate reasoning approaches that repeatedly access and manipulate different scene elements, as dictated by complex intermediate computational or scene graphs (Fei et al., 2024; Ji et al., 2020). Furthermore, when these reasoning models are scaled, such as using Vision-Language Models (VLMs), they require even more compute and GPU memory (Bordes et al., 2024). Hence, given a limited compute budget, VLMs only operate on a few images or short snippets/summaries and often struggle to efficiently perform dense understanding of videos, limiting video sizes to a few minutes (Weng et al., 2024).

Even though several efficient approaches exist, these methods often sample a fixed number of frames (Ma et al., 2018) affecting model performance for certain actions, or perform a clip based aggregation losing the order of short term actions (Fan et al., 2021a). Existing token sampling and pruning methods condense background tokens in the spatial domain, and do not store or re-use tokens in memory that can affect efficiency for dense spatio-temporal tasks (Bolya et al., 2022; Fayyaz et al., 2022).

Figure 1a illustrates the computational challenge associated with video reasoning models. As shown in Figure 1(a), TubeDETR (Yang et al., 2022) repeatedly processes a large number of frames while handling approximately 6000 activity queries on long videos during inference. The average duration of these long videos spans is 27 minutes. A histogram of the frame processing counts is presented in Figure 1a. This observation demonstrates an opportunity for memory-based approaches, which load the frames of long videos into GPU memory, to improve computational efficiency.

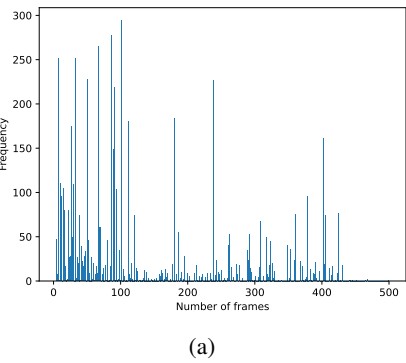

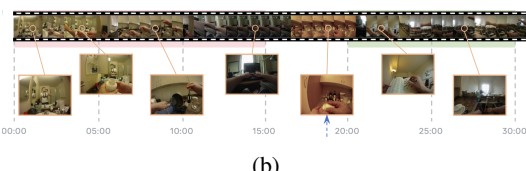

(a)                                           (b)

Figure 1: Figure 1(a) shows a activity query video and a histogram of the number of times a single frame is reloaded in GPU memory for the video from the video reasoning, ReST-ADL dataset (FPS=1). (b) Activity video. Image source (Yang et al., 2023)

To address the above issues, we present LVM-Net which uses a fixed-size memory representation to identify and store discriminative patches sampled from the input video. The memory patches are identified using a *neural* sampler, that improves efficiency while maintaining the discriminability of memory representation. Additionally, LVM-Net only requires a single pass of the video over the memory, further improving the overall efficiency.

In our results over the Rest-ADL dataset, we demonstrate an 18X speedup during inference with 1 FPS and and 75x improvement with 5 FPS, for long form video retrieval for answering questions over long (>30 min) videos to answer activity, object, and temporal queries and achieve competitive performance.

## 2 RELATED WORK

Our work is inspired by related work in video understanding methods including long-form video understanding, reasoning and efficient video transformer architectures.

**Video understanding**  Deep learning based video understanding methods have evolved from using 3D convolution based methods (Ji et al., 2012) to 2D-CNNs (Donahue et al., 2015; Simonyan & Zisserman, 2014; Feichtenhofer et al., 2019), with additional blocks such as object/ROI features (Gkioxari et al., 2018; Ma et al., 2018), convolution-transformer approaches (Girdhar et al., 2019) and transformer-only approaches (Arnab et al., 2021; Bertasius et al., 2021; Fan et al., 2021b; Liu et al., 2022). Transformer based approaches often tokenize the video by chopping the input into a series of 2D spatial frames or into 3D spatio-temporal cubes on a regular grid. This approach can provide high accuracy but requires significant amounts of compute and memory due to large number of tokens and their parallel processing in transformer architecture. In contrast, our method uses transformer based tokens and samples the tokens to significantly reduce processing costs.

**Long-form video reasoning**  Various long-form video understanding approaches have been studied  (Song et al., 2024; Sun et al., 2022; Wang et al., 2024; Wu & Krahenbuhl, 2021; Wu et al., 2022). Noteworthy among these approaches, MeMViT caches representations of previous clips to extend temporal context without increasing per-step compute. However, these approaches are often limited to less than few minutes, largely due to lack of long form video datasets (> 30 mins). Additionally, existing reasoning based datasets largely focus around QA (Yi et al., 2019) or temporal action retrieval (Chao et al., 2018; Hahn et al., 2019; Yuan et al., 2016). In contrast, our paper is focused on videos with duration of 30+ mins with a focus on tasks that require a joint analysis of activities, objects and time, which requires complex reasoning.

**Long-context VLMs**  Most VLMs can usually process only a few minutes of videos due to limited context length. Video processing requires a large number of tokens to be processed; for example, deploying a 7B Llama model that can process 10 million tokens requires 8 A100 GPUs (640GB

memory), even with advanced serving optimizations (Hooper et al., 2024). Even larger proprietary models such as Gemini 1.5 Pro can process 10 million tokens which roughly translates to approximately 10 hours of video duration(Reid et al., 2024). Gemini 1.5 model architecture and number of parameters are unknown. However, Gemini 1.5 model is most likely compute intensive – based on its API pricing (GeminiAPI, 2024). In contrast, our proposed LVM-Net consists of around 300 million parameters ($\leq 1$ GB with FP16) and can process a single 10 hour video into a fixed sized memory ($\leq 1$ GB with FP16). LVM-Net can be deployed on an edge device.

**Efficient transformer architectures** Efficient transformer architectures have focused on reducing the cost of quadratic attention costs with respect to sequence lengths (Tay et al., 2020), pruning (Meng et al., 2022; Rao et al., 2021; Voita et al., 2019) and reduction of vision tokens as an input to decoders downstream. Previous work has analyzed sparse attention patterns to reduce complexity from attention to linear (Beltagy et al., 2020; Zaheer et al., 2020), and approximated attention using kernel methods achieving linear time and memory complexity (Choromanski et al., 2020; Schlag et al., 2021). Many hierarchical approaches use a hierarchical token structure to process inputs at multiple resolutions, reducing overall computation (Jaegle et al., 2021; Liu et al., 2021a; Feng et al., 2023).

**Token efficiency methods** In order to reduce token costs, approaches such as token merging (Bolya et al., 2022), adaptive token sampling in classification domain (Fayyaz et al., 2022), token turing machines (Ryoo et al., 2023), spatio-temporal token selection (Wang et al., 2022) have been explored. BLIP-3 (Xue et al., 2024) uses Perceiver based token sampler to project input image to a fixed number of tokens. However, token pruning methods often can deduplicate tokens whereas, LVM-Net identifies discriminative tokens using a neural sampler. Crucially, LVM-Net uses a fixed memory, re-using token representations across queries significantly reducing inference time.

## 3 PRELIMINARIES

We use the Relational Space-Time Query (ReST) dataset (Yang et al., 2023) to evaluate long-form video reasoning. ReST consists of three kinds of relational space-time queries: activity query, object query, and time-query. Each query asks questions on a single property (e.g. activity) by providing the other two properties (e.g. object and time) as input.

The templates of queries are as follows:

1. Activity query - what *activities* did I perform with a particular *object* during a given *time*?

2. Object query - on which *objects* I perform with a particular *activity* during a given *time*?

3. Time query - at what *time* did I perform a particular *activity* with a particular *object*?

ReST consists of long videos with average duration of 27 minutes in length. The relational space-time queries over the videos are further categorized into three types based on query time duration – short (around 5 minutes), medium (around 15 minutes), and long (around 30 minutes). Note that the short queries in our paper are longer than those typically employed by existing models, which usually last about 3 minutes (Yang et al., 2023).

There are four-time representations in our setup: long video time $(v_s, v_e)$, ReST query time $(q_s, q_e)$, time-property time $(t_s, t_e)$ and clip time $(c_s, c_e)$. The long video time represents the complete duration of the input video clip (up to an hour long). The ReST query time $(q_s, q_e)$ represents the duration of the relational space time query. The time duration of a ReST query can be short (around 5 mins), medium (around 15 minutes), and long (around 30 minutes). The ReST query time has the following constraint: $v_s \leq q_s \leq q_e \leq v_e$. The time-property time $(t_s, t_e)$ of a query is the duration when an *activity* occurs on an *object* within query time $(q_s, q_e)$. The time-property time has following constraint: $v_s \leq q_s \leq t_s \leq t_e \leq q_e \leq v_e$. The clip time represents the sampled clip from a long video such that the sampled frames from the clip can be loaded into the available GPU memory. It has following constraint: $v_s \leq c_s \leq c_e \leq v_e$. The ReST dataset contains multiple queries per video where $q_j^i$ represents the $j^{th}$ query in the ReST dataset that belongs to video $i$.

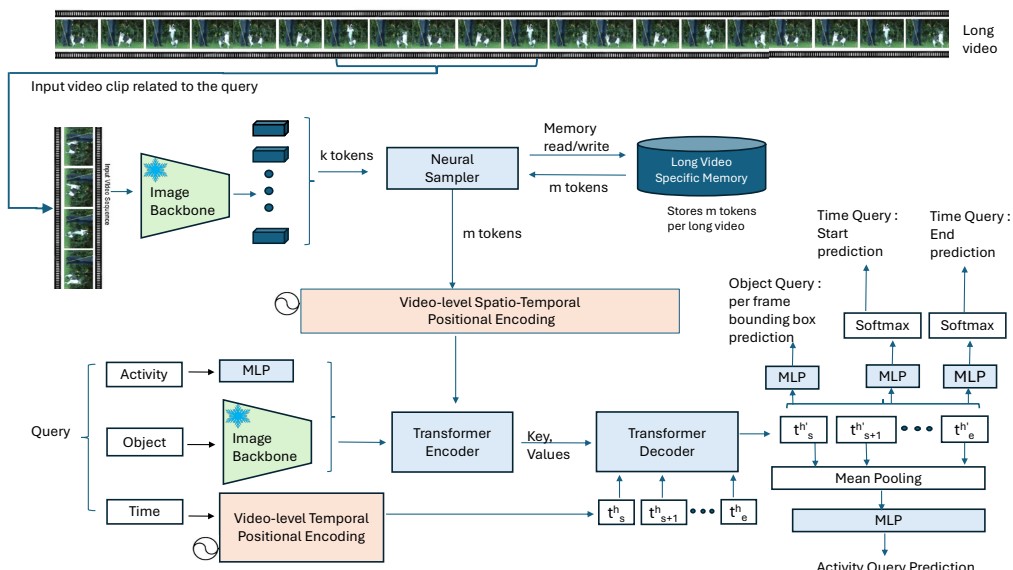

Figure 2: **Overview of the LVM-Net training : We sample tokens from input videos and store them in memory to efficiently process long-form videos. The inference steps are shown in the Figure 3.**

# 4 METHODOLOGY

We now describe the model architecture for efficient video reasoning. We refer to video reasoning as a model's ability to understand three properties: what *activity* is being performed on what *object* over what *time*. We test a model's video understanding ability by asking queries where we provide input two properties of the video and then ask the model to predict the third property.

In order to build an efficient architecture, we draw inspiration from human memory that uses multiple memory representations and uses attention as a gatekeeper for the memory, guided by the high level goals (Hazy et al., 2006; Watzl, 2017). LVM-Net performs reasoning over long videos by using attention to store specific information within a fixed memory. It achieves this through a trained neural sampler that extracts discriminative visual patches from the video and stores them in memory as shown in Figure 2. During inference, LVM-Net uses this pre-populated memory to respond to queries without needing to revisit the original video, significantly reducing inference time. This architecture is well-suited for answering multiple queries from a single video, enabling fast and effective video understanding.

LVM-NET INPUT ENCODING

The three properties in ReST—*activity*, *object*, and *time*—are represented in three different representation spaces: activity is represented as one of $\mathcal{C}$ classes, the object is represented as an image, and time is represented by start and end times.

The *activity* input is represented as a one-dimensional vector $a_j \in \mathbb{R}^{1 \times \mathcal{C}}$. This vector is then passed through a feed-forward layer to obtain a $d$-dimensional representation $a_j^h \in \mathbb{R}^{1 \times d}$. The *object* input is an instance image $o_j \in \mathbb{R}^{o_h \times o_w}$. The image is passed through a frozen image backbone (pretrained Swin Transformer (Liu et al., 2021b)). The *time* input $(t_{j,s}, t_{j,e})$ from the ReST query is passed into a video-level temporal positional encoding layer (Vaswani et al., 2017) that outputs a latent representation $t_j^h \in \mathbb{R}^{(t_{j,e}-t_{j,s}+1) \times d}$.

## NEURAL SAMPLER

The objective of the differentiable neural sampler is to populate memory with the representative visual tokens from the whole long video $v_i$. The memory stores a fixed number of $m_i \in \mathbb{R}^{m \times d}$ visual tokens per video. The input to the neural sampler is $m_i$ memory tokens and $k$ visual tokens from the sampled clip $c_j^i \in \mathbb{R}^{T \times H \times W \times d}$ of the query $q_j^i$ where $T, H$, and $W$ are duration of clip, height and weight of the image frames, respectively. The neural sampler outputs $m$ discriminative visual tokens. The sampled clip $c_j^i \in \mathbb{R}^{T \times H \times W \times d}$ is passed through a pre-trained, frozen Swin transformer based image backbone (Liu et al., 2021b) that results in $k \in \mathbb{R}^{Th'w' \times d}$ visual tokens where $h'$ and $w'$ are computed based on patch size. We use the same image backbone for object image and clip frames. The sampled $m_i$ tokens are then passed through the 2D spatial positional encoding layer and video-level temporal positional encoding layer. Our proposed framework is independent of the choice of neural sampler (Xie et al., 2019; Pervez et al., 2022).

The neural sampler outputs scores for $k$ (clip) tokens and $m$ (memory) tokens. To understand which tokens out of $m + k$ tokens are important, we first pass the $m + k$ tokens through a transformer encoder followed by a single MLP layer that outputs the scores. The neural sampler (Xie et al., 2019) samples subsets with Gumbel-Top $k$ Relaxations that adds Gumbel noise to the scores and utilizes reparameterization trick (Kingma, 2013) so that the gradients can back-propagate to the transformer encoder and MLP layer. The sampler is trained based on ReST queries predictive performance where the loss is higher if the sampler samples non-discriminative tokens.

## ENCODER-DECODER

The input $x_j^i$ to the transformer encoder includes $m_i \in \mathbb{R}^{m \times d}$ memory tokens along with the query specific input. For example, in the case of activity query, the input includes latent representation of instance image $o_j^h$ so the input is $x_j \in \mathbb{R}^{(m+oh'ow') \times d}$. Before passing the input $x_j$ to the encoder, we perform element-wise multiplication of instance image and frame tokens. Let $nf$ be number of frames, then $m = nf \times (oh'ow')$. Therefore, $x_j \in \mathbb{R}^{((nf+1) \odot oh'ow') \times d}$. In case of object query, the input includes latent representation of activity $a_j^h$ so input is and $x_j \in \mathbb{R}^{(m+1) \times d}$. In case of time query, the input includes both latent representation of activity $a_j^h$ and instance image $o_j^h$ so input is $x_j \in \mathbb{R}^{(m+oh'ow'+1) \times d}$ and after element-wise multiplication $x_j \in \mathbb{R}^{(((nf+1) \odot oh'ow')+1) \times d}$. The transformer decoder accepts input in the form of key and value representations from the transformer encoder. The queries input to the transformer decoder are initialized based on video-level temporal positional encoding, with an input given by $t_j^h \in \mathbb{R}^{(t_{j,e}-t_{j,s}+1) \times d}$. The output of the decoder is a learned representation of $\hat{t}_j^h$, which are learned by contextualizing memory tokens and the ReST queries' input representations through the use of time query representations.

## LVM-NET OUTPUT ENCODING

The learned representation of $\hat{t}_j^h$ from the previous step is used for prediction tasks. The prediction is carried out using a query-specific multi-layer perceptron (MLP) head. For an activity query, we apply mean pooling to $\hat{t}_j^h \in \mathbb{R}^{(t_{j,e}-t_{j,s}+1) \times d}$ to obtain $\hat{t}' \in \mathbb{R}^{1 \times d}$. This representation is then fed into an activity prediction MLP, which predicts the activities $\hat{a} \in \mathbb{R}^{1 \times C}$.

For an object query, bounding box predictions are computed for each sampled frame. To do this, we pass the learned query representation $\hat{t}_j^h \in \mathbb{R}^{(t_{j,e}-t_{j,s}+1) \times d}$ through a specific MLP layer tailored for object queries. This object-specific MLP layer predicts normalized bounding boxes $\hat{o}_j \in \mathbb{R}^{(t_{j,e}-t_{j,s}+1) \times 4}$ for each sampled frame.

In the case of a time query, we pass the learned query representation $\hat{t}_j^h \in \mathbb{R}^{(t_{j,e}-t_{j,s}+1) \times d}$ through two separate MLP layers to predict start and end times.

## MEMORY READ/WRITE OPERATIONS

In LVM-Net the memory $m_i$ is allocated per long video $v_i$. A long video $v_i$ can possibly have multiple associated ReST queries where each query could focus on a clip (for example, $c_j^i$ which

corresponds to $j^{th}$ clip of video $v_i$). Since we train our model through sampled clips it becomes important, how we form a batch through a data sampler. A naive data sampler can form a batch with two or more clips belonging to the same video. In this case, the neural sampler would read $m$ tokens from video $v_i$ and $k$ tokens from each clip $c_j^i$ (say $c_{j1}^i$ and $c_{j2}^i$ ). The neural sampler would then output $m$ tokens for $c_{j1}^i$ and $m$ tokens for $c_{j2}^i$ to be written to the $i^{th}$ video memory slot thereby creating a race condition [1].

To avoid this race condition on a single GPU, we design a data sampler such that a batch has ReST queries with no two queries belonging to the same long video. In distributed training with multiple GPUs, our data sampler ensures that all the batches have ReST queries with no two queries belonging to the same video. This data-sampling constraint ensures there is no memory corruption. At the end of each iteration, the written memory tokens are synchronized across all devices. To summarize, in a distributed training setup with $r$ devices and $n$ number of videos, the maximum batch size on a single GPU becomes $\frac{n}{r}$ in order to avoid race condition.

### INFERENCE

The inference of LVM-Net enables faster processing of queries than existing long video understanding models. In existing models such as TubeDETR (Yang et al., 2022), for answering $q$ queries from a single video, one has to pass the query clip's frames, $q$ number of times. The clip processing – passing the clip frames through the image backbone and then passing the latent representations through encoder-decoder modules – is compute intensive and results in a significant delay in generating the query's response. Moreover, $q$ queries are processed independently so if multiple queries share a small region of clip, there is no potential to offset the clip processing load. In contrast, in our proposed model, as shown in the Figure 3, we first populate video $v_i$ specific memory $m_i$ through our trained neural sampler. All the responses to the queries that belong to video $v_i$ are generated using sampled memory $m_i$.

The memory $m_i \in \mathbb{R}^{(m,d)}$ – $m$ is the number of tokens and $d$ represents latent dimension of image backbone – for a particular video $v_i$ is populated as follows: we first initialize $m_i$ with video tokens sampled randomly. We then extract clips from video $v_i$ through a sliding window with two clips having zero overlap. Each clip is then passed through the image backbone that outputs $k$ tokens. The neural sampler takes $m$ memory tokens and $k$ clip tokens and outputs $m$ tokens that are written to memory. In the end, the populated $m_i$ is fed to the encoder for answering queries that belong to $i^{th}$ video.

An additional advantage of our proposed model is that it can be deployed on an edge device with limited memory. The inference can be performed in a streaming fashion where we can store $m$ memory tokens and $k$ tokens from the current query clip. The sampler here would take input $m + k$ tokens and output $m$ discriminative tokens. These $m$ memory tokens are used to generate responses for multiple queries.

### TRAINING LOSS

The input training data is the ReST query's clip frames. In the case of activity query, the input is two properties: object instance and time-property time $(t_s, t_e)$. The task in an activity query is to predict the activity from the available $C$ classes. In an activity query, multiple activities can happen on an object instance within time-property time, so we model the activity prediction as a multi-label classification setup. We use focal loss $(a, \hat{a})$ where $a \in \mathbb{R}^{1 \times C}$ represents ground truth.

In the case of object query, the input is two properties: activity class and time-property time $(t_s, t_e)$. The task in the object query is to predict bounding boxes for each sampled frame in $(t_s, t_e)$. Given ground truth bounding boxes, $o$ where $o \in [0, 1]^{4(t_e - t_s + 1)}$ and predicted bounding boxes $\hat{o}$, the object query loss is given as

$$\sum_{i \in \text{object-queries}} \lambda_1 \mathcal{L}_1(\hat{o_j}, o_j) + \lambda_{gIoU} \mathcal{L}_{gIoU}(\hat{o_j}, o_j) \tag{1}$$

---

[1] A race condition occurs where two or more processes attempt to write to the same shared memory at the same time.

|  | Activity Query | Object Query | Time Query |
|---|---|---|---|
| **Short Queries** | | | |
| Modified TubeDETR | 264 mins | 99 mins | 11 mins |
| *LVM-Net* | 14 mins (18x) | 6 mins (16.5x) | 7 mins |
| **Medium Queries** | | | |
| Modified TubeDETR | 180 mins | 663 mins | 31 mins |
| *LVM-Net* | 16 mins (11.2x) | 15 mins (44x) | 14 mins |
| **Long Queries** | | | |
| Modified TubeDETR | 174 mins | 756 mins | 19 mins |
| *LVM-Net* | 15 mins (11.6x) | 10 mins (75x) | 10 mins |

Table 1: **Running time: Benchmarked over a single A100 with inference batch size selected to maximize 80GB GPU memory for both methods. The Target FPS is set to one for activity and time query and set to five for object query as ground truth is available at five FPS for object query.**

where $\mathcal{L}_1$ is $\mathcal{L}_1$ loss on bounding boxes coordinates and $\mathcal{L}_{gIoU}$ is generalized intersection over union loss on the bounding boxes (Rezatofighi et al., 2019). $\lambda_1$ and $\lambda_{gIoU}$ are scalar weights.

In the case of time query, the input is three properties: activity class, object instance, and query time $(qt_s, qt_e)$. The task in time query is to predict the time-property time $(t_s, t_e)$ within $(qt_s, qt_e)$. The ground truth is represented through two vectors – $v_{t_s} \in \mathbb{R}^{1 \times l}$ for start time $t_s$ and $v_{t_e} \in \mathbb{R}^{1 \times l}$ for end time $t_e$. Here, $l$ is set to $(t_e - t_s)/$target-fps. We compute Cross Entropy loss $\mathcal{L}_{CE}(\hat{v}_{t_s}, v_{t_s}) + \mathcal{L}_{CE}(\hat{v}_{t_e}, v_{t_e})$ for training the model on time query.

### ONLINE CONTINUAL LEARNING LOSS

Given a ReST query $q_j^i$, $k$ visual tokens from the $q_j^i$'s clip, and $m_i$ memory tokens of video $i$, we train the neural sampler based on the training loss. However, with this training setup, the neural sampler is biased towards sampling $q_j^i$'s clip visual tokens instead of memory tokens – since the training loss computed on $q_j^i$'s predictions is minimized by sampling visual tokens from the $q_j^i$'s clip. As a result, the model cannot identify tokens that capture the global view of the long video. This bias contrasts against our goal of training a neural sampler that would process the long video *once* and populate discriminative tokens into memory.

We propose an auxiliary loss to address this sampling bias. Specifically, we propose online continual learning loss shown in Figure 4. Here, we store past $p$ ReST queries in a heap of size $p$ where the oldest query is ejected when the heap is full. These ReST queries are passed through the shared transformer encoder-decoder and the training loss is computed on both the current query's predictions and past $p$ query's predictions. This auxiliary loss addresses the sampling bias of the neural sampler. We make a design choice of performing the continual learning in an online fashion – training based on recent past $p$ queries – instead of randomly sampling $p$ queries from all the previous queries. This is due to the fact that the initial probability of past $p$ queries' relevant tokens being in memory is high. With online continual learning, we reinforce the neural sampler to give those relevant tokens high scores regardless of their relevance to the current query $q_j^i$.

### EXPERIMENTS

We performed experiments on the ReST-ADL dataset. The ReST-ADL consists of three relational space-time queries – activity, object, and time. The train and test splits are performed at the level of individual videos. Each ReST query is evaluated on three different query durations: short (around 5 mins), medium (around 15 mins), and long (around 30 mins).

|  | Activity Query | Object Query | Time Query |
|---|---|---|---|
| **Short Queries** | | | |
| ReST system | 48.1 | 9.6 | 31.3 |
| Modified TubeDETR | 45.3 | 27.5 | 35.0 |
| *LVM-Net* | 32.4 | 26.4 | 22.9 |
| **Medium Queries** | | | |
| ReST system | 50.7 | 10.0 | 31.8 |
| Modified TubeDETR | 31.6 | 25.4 | 6.7 |
| *LVM-Net* | 26.1 | 11.9 | 11.9 |
| **Long Queries** | | | |
| ReST system | 46.3 | 10.0 | 30.0 |
| Modified TubeDETR | 29.9 | 24.6 | 12.8 |
| *LVM-Net* | 22.8 | 21.3 | 8.6 |

Table 2: **Prediction Performance (Recall@1x) over short, medium, and long queries using ReST, TubeDETR, and LVM-Net. We demonstrate that LVM-Net achieves competitive performance despite the 18X speedup in inference time.**

EVALUATION METRICS

We follow (Yang et al., 2023) and use recall@1x metric for evaluation. The metric measures the percentage of ground truth labels identified in top $x$ predictions where $x$ stands for the number of ground truth predictions. In case of object query, we follow (Yang et al., 2022) and define $vIoU_j = \frac{1}{S_u} \sum_{f \in t_e - t_s + 1} IoU(\hat{o}_{j,f}, o_{j,f})$. The prediction is positive if $vIoU_j > R$ otherwise a zero value is assigned to the prediction. Following (Yang et al., 2022), we set $R = 0.3$. In the case of time query, we again compute $tIoU$ using ground truth start-end time and predicted start-end time. A prediction is positive if $tIoU_j > 0.3$ otherwise a zero value is assigned to the prediction.

BASELINES

We compare our proposed method with the ReST (Yang et al., 2023) that uses a multi-stage differentiable learning model and end-to-end TubeDETR method (Yang et al., 2022). We modify the last MLP layer of TubeDETR for activity prediction outputs. TubeDETR operates on clips that can be loaded into GPU memory. For clips with durations greater than 4 minutes (1 FPS), TubeDETR requires sampling a fixed number of frames to meet GPU memory requirements. We follow the clip-based training and inference recipe outlined in the PyTorchVideo library (Fan et al., 2021a) for TubeDETR. Specifically, during training, given a long clip, we randomly sample a sub-clip whose duration, with the selected FPS, results in a predefined fixed number of frames. During inference, we follow these steps:

1. Divide the long clip into non-overlapping short clips.

2. Pass each short clip along with the object image through the trained TubeDETR model, which outputs the activity class logits.

3. Aggregate the logits across all clips and perform prediction.

In the case of the object query, during training, we apply object detection loss mentioned in the Equation 1 on the sampled clip. During inference, we divide the long clip of query $q_j$ into non-overlapping short clips, pass the activity hidden representation along with each clip and compute the $vIoU_j$ score per query.

In the case of the time query, where the task requires predicting the start and end times, we follow TubeDETR and sample a fixed number of frames (Yang et al., 2022).

RESULTS

We report our experimental results in Tables 1 and 2. We perform the inference running time computation on the identical A100 instances. Batch size for both the methods is selected to maximize the GPU memory utilization. As shown in Table 1, LVM-Net, outperforms the TubeDETR model in terms of inference speed. In the case of activity query, LVM-Net achieves speedups of 18X, 11.2X, and 11.6X over TubeDETR on short, medium, and long activity queries, respectively. When processing the ReST-ADL dataset, which consists of approximately 6000 test activity queries across four long videos, LVM-Net passes each video through its neural sampler once to create four video-specific memories. All subsequent test queries are then processed using this memory, resulting in efficiency gains.

In contrast, TubeDETR treats each query independently and requires a separate inference process for every query, as outlined in the baselines section. This approach leads to redundant processing of frames when multiple queries refer to overlapping or identical long clips. While it might be possible to optimize this by processing all frame representations once and storing them on disk, this approach would still require additional steps:

1. Dividing the long video clip into non-overlapping short clips.

2. Loading these clips' frame representations in memory.

3. Passing them along with the object image through the trained TubeDETR model to obtain logits.

4. Aggregating these logits.

The latter steps are particularly time-consuming and result in slow inference times.

In the case of object query, the ground truth is available at five FPS, hence the target fps is set to five for all the models as the groundtruth for object query is available at five FPS. We observe 75x improvement in inference speed as compared to TubeDETR since at five FPS, TubeDETR has to process 5X more frames. From Table 2, we observe LVM-Net performs competitively as compared to TubeDETR.

In the case of the time query, due to the nature of predicting start and end times, we follow Tube-DETR and sample a fixed number of frames(Yang et al., 2022). We observe that the modified Tube-DETR had a shorter running time for time queries compared to activity and object queries due to frame sampling. However, for medium (15-minute) and long (30-minute) queries, the performance of the modified TubeDETR deteriorates because it is not explicitly designed for long videos(Yang et al., 2022). In contrast, LVM-Net stores a global view of each video in memory and passes this along with object images through its trained encoder-decoder model, which outputs activity class logits without requiring any additional aggregation. As shown in Table 2, our system performs competitively compared to other methods.

We also perform additional experiments (reported in the appendix) where we demonstrate the impact of online continual vs non-continual learning loss in Section A.3.2. We also measure the trained neural sampler's ability to sample discriminative tokens by comparing the performance of LVM-Net with trained neural sampler vs uniform random sampling of tokens in Section A.3.1.

CONCLUSION

Long-form video understanding has been a long-standing challenge for the computer vision community. While many approaches exist, they cannot be efficiently applied to longer videos over 30 minutes. In this paper, we present LVM-Net that demonstrates an efficient network for long-form video reasoning using an external memory. The external memory is populated using differentiable neural sampler that samples tokens and builds an effective condensed representation. In our results, we demonstrate an 18-75X faster inference over the state of the art in the ReST ADL video reasoning benchmark. The inference speedup is primarily due to the fact that our proposed LVM-Net performs a single pass over a long video to populate memory and can provide answers to multiple queries from the long video using the populated memory. In the future, we plan to extend our work to long-context VLMs and to reduce the compute requirements for retrievals using natural language.

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

# A  APPENDIX

## A.1  INFERENCE

The inference pipeline is shown in Figure 3. In the first stage, the memory is first initialized with random video tokens. We then divide the long video into multiple non-overlapping clips. These clips are then passed through the trained neural sampler in a random order. One can also pass these non-overlapping clips multiple times through the neural sampler. However, we observe minor improvement in the performance. The populated video specific memory $m_i$ is then utilized to provide responses to ReST queries $q_j^i$.

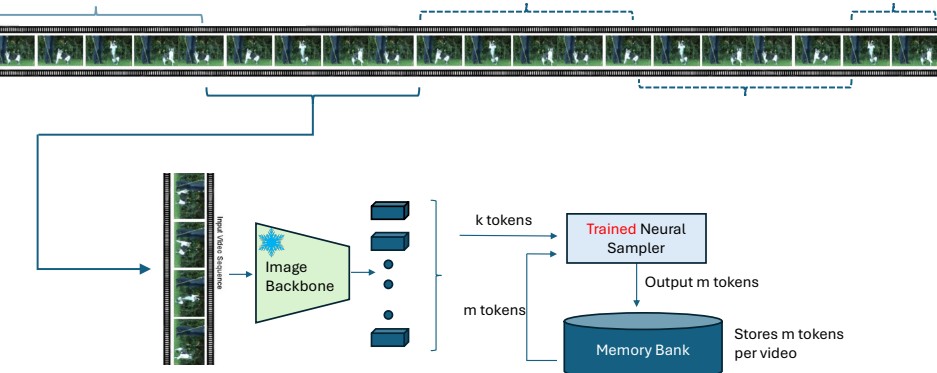

(a) Inference Stage 1: The whole long video is passed clip by clip through the trained neural sampler which populates the memory.

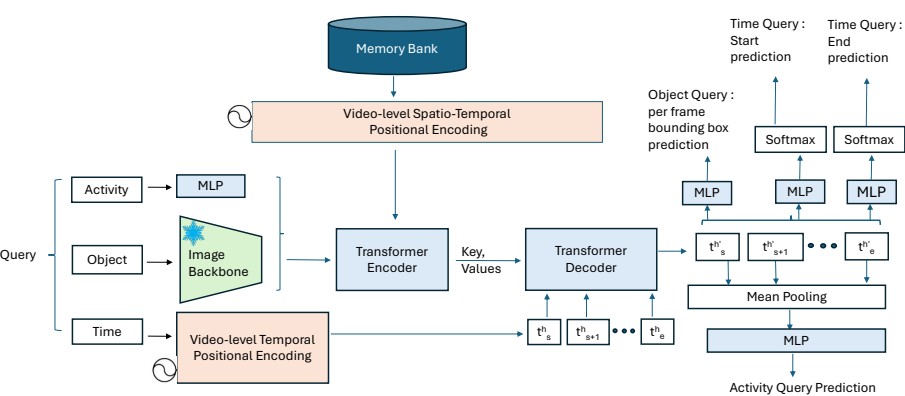

(b) Inference Stage 2: The ReST queries responses are predicted by our trained model by only reviewing the pre-computed memory tokens.

Figure 3: Two stage Inference pipeline of LVM-Net .

## A.2  ONLINE CONTINUAL LEARNING

The continual learning loss is shown in Figure 4. We store the past $p$ number of ReST queries in a heap of size $p$ where the oldest query is ejected when the heap is full. The $p$ ReST queries when passed through LVM-Net output $p$ loss values. For $p > 1$, we compute the sum of those $p$ losses and add it to the current query's loss.

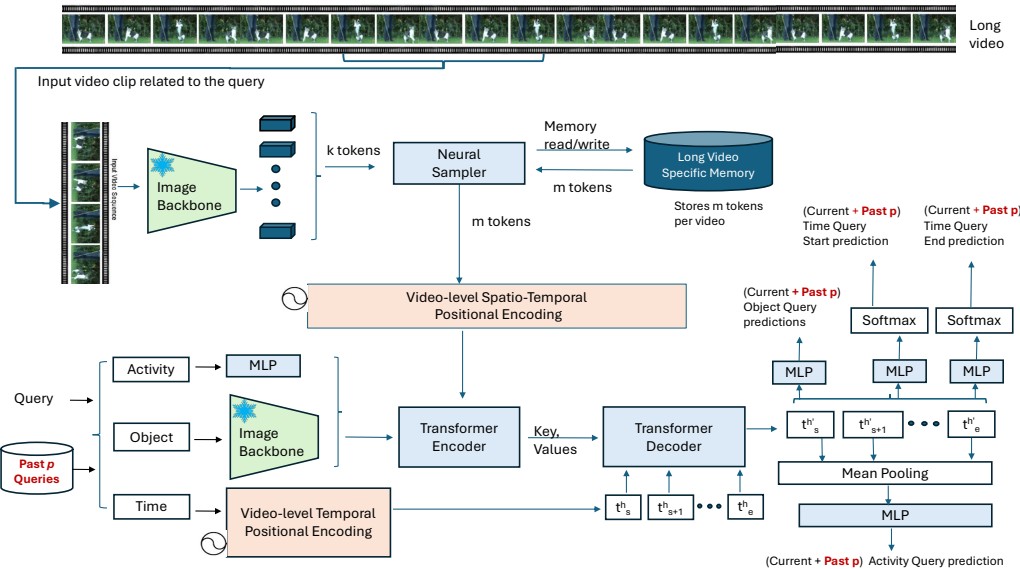

Figure 4: **Auxillary online continual learning loss (shown in red color). The loss addresses the bias of neural sampler towards sampling current query's clip tokens rather than sampling tokens that helps reduce loss for all the queries.**

| | Recall@1x | Recall@3x |
|---|---|---|
| **Short Queries** | | |
| *LVM-Net* | 32.38 | 56.78 |
| *LVM-Net*-random | 21.42 | 43.20 |
| **Medium Queries** | | |
| *LVM-Net* | 26.12 | 44.80 |
| *LVM-Net*-random | 18.23 | 40.57 |
| **Long Queries** | | |
| *LVM-Net* | 22.81 | 45.39 |
| *LVM-Net*-random | 18.42 | 38.45 |

Table 3: **Activity Query: Neural sampler vs uniform random of tokens**

## A.3 ABLATION STUDIES

### A.3.1 RANDOM SAMPLING VIDEO TOKENS VS SAMPLER

We study the impact of the neural sampler in sampling video tokens as compared to the uniform sampling of tokens in Table 3. We quantitatively show that the neural sampler is able to identify discriminative tokens as compared to a uniform random sampling of tokens. The uniform random sampling would sample a lot of background tokens as compared to the trained neural sampler thereby resulting in a significant reduction in the predictive performance of activity query.

|  | Recall@1x | Recall@3x |
|---|---|---|
| **Short Queries** | | |
| *LVM-Net* | 32.38 | 56.78 |
| *LVM-Net*-non-continual | 26.39 | 47.31 |
| **Medium Queries** | | |
| *LVM-Net* | 26.12 | 44.80 |
| *LVM-Net*-non-continual | 24.81 | 44.63 |
| **Long Queries** | | |
| *LVM-Net* | 22.81 | 45.39 |
| *LVM-Net*-non-continual | 18.28 | 44.54 |

Table 4: **Activity Query: Continual Learning vs Non-Continual learning**

### A.3.2 CONTINUAL LEARNING

We perform an ablation experiment where we report the performance of LVM-Net with and without continual learning in Table 4. We can see that adding continual learning helps improve the performance of LVM-Net in a significant manner.

### A.4 LVM-NET DETAILS

The temporal positional encoding layer is standard positional encoding (Vaswani et al., 2017) where the sequence length is set to the maximum long-video length in seconds times the target FPS. We set the target frame per second (FPS) to 1 for activity and time query while the target FPS is set to 5 for object query. We sample 120 number of frames set in a clip. We select the frozen pre-trained image backbone as Swin transformer (Liu et al., 2021b). We set the following hyper-parameters: $T = 120, d = 2048, N = 2, \mathcal{L}_1 = 5, \lambda_{gIoU} = 2$. We train our model and baseline models for 10 epochs. The models are trained on 4 A100 GPUs with an effective batch size of 4. We initialize the parameters of LVM-Net using modified TubeDETR. The learning rate of the neural sampler is set to $1e$-5 while the rest of the parameters learning rate is set to $1e$-7. We reset the memory bank after every training epoch. We set the number of past continual learning queries $p$ value to 2. The memory size is set to 5880 tokens. The Swin transformer outputs 49 tokens per frame. With 120 number of frames, the number of clip tokens and memory tokens has the same capacity of tokens. The number of layers in all MLPs is set to 1. We use the TIMM library Wightman (2019) for Swin transformer backbone with model id: swinv2_cr_small_ns_224. We perform data augmentations – horizontal flip, posterize, photometric distortion – with a probability of 0.25. The dropout value is set 0.2. To encourage exploration during the initial stage of neural sampler training, we set the temperature to 1.5 and slowly decrease the value of the temperature to 1.

