# OpenReview forum: "LVM-NET: Efficient Long-Form Video Reasoning using neural sampling"
_ICLR.cc/2025/Conference — Submitted to ICLR 2025_

### Official Review · Reviewer_HJDb · 2024-10-30

**Soundness:** 3
**Presentation:** 3
**Contribution:** 3
**Rating:** 3
**Confidence:** 5

**Summary:**

The authors propose LVM-Net training, with a Long Video Specific Memory that

**Strengths:**

1. The proposed neural sampler is easy to follow.

**Weaknesses:**

1. This article is very poorly written with numerous grammar errors. For example, lines 152-161、420-425 and 448-453 are prolixity. The author consistently omits the articles "a," "an," and "the." I suggest the author revise the manuscript carefully.

2. Line 011 and Line 025 are contradictory. Is it reasoning or understanding? The terms `video reasoning` and `video understanding` are often used interchangeably in this paper, but they have different meanings. `Reasoning` is the process of using logic to draw conclusions, while `understanding` is the comprehension and interpretation of information.

3. What's the motivation for using the neural sampler?

4. Aside from the neural samples, what are the main differences between this paper and the previous work TubeDETR?

5. This paper only reports the performance in the ReST-ADL dataset. The authors may consider reporting the performance on other datasets to demonstrate the generalizability.

**Questions:**

1. What's the purpose of Figure 1(b)? which is not discussed in this paper.

---

### Official Review · Reviewer_FyyA · 2024-10-30

**Soundness:** 2
**Presentation:** 1
**Contribution:** 2
**Rating:** 3
**Confidence:** 4

**Summary:**

To solve the computational requirements and memory constraints in long video understanding, the authors develop LVM-Net, which first selects discriminative memory tokens, and stores in a fixed length memory bank. With this approach, LVM-Net achieves significant efficienct improvement on Rest-ADL dataset, along with a competitive predictive performance.

**Strengths:**

1. **Innovative Efficiency in Video Reasoning**: The paper introduces LVM-Net, a novel method that significantly reduces computational demands and GPU memory usage by using a fixed-size memory representation and a neural sampler to store discriminative patches from the input video, which requires only a single pass over the video data
2. **Significant Performance Improvements**: Demonstrates substantial inference speed-ups ranging from 18x to 75x on the Rest-ADL dataset for long-form video retrieval and question answering tasks, achieving these gains without compromising predictive performance.

**Weaknesses:**

1. **Incomplete Review of Recent Works:** The paper’s introduction of recent works is not comprehensive enough. For example, although the Introduction states, “Existing token sampling and pruning methods condense background tokens in the spatial domain, and do not store or re-use tokens in memory that can affect efficiency for dense spatio-temporal tasks,” it lacks a thorough discussion of related methods and does not cite relevant literature that addresses these issues.

2.	**Typos and Inconsistencies:** There are several typos and inconsistencies in the paper. For instance, when referring to Figure 1(a), it is sometimes written as “Figure 1a.” The notation should be consistent throughout the paper to avoid confusion.

3.	**Missing Figure:** The caption of Figure 2 mentions that “The inference steps are shown in Figure 3,” but Figure 3 is missing from the paper.

4.	**Unclear Method Description:** The explanation of the proposed method is confusing. It is unclear which parts of the model require training—is it only the neural sampler, or does it also include the positional encoding layer?

5.	**Incomplete Inference Time Reporting:** In Table 2, the authors compare the performance of ReST, TubeDETR, and LVM-Net. However, in Table 1, they report the inference time of Modified TubeDETR but not ReST. I am interested in knowing the inference time of the ReST system for a fair comparison.

6.	**Redundant Introduction of the ReST Dataset**: The paper introduces the ReST dataset multiple times, including in the preliminaries section, which seems unnecessary.

7.	** Limited Scope Despite Broad Claims**: Although the authors state in the abstract that “Long-form video reasoning is essential for various applications such as video retrieval, summarizing, and question answering,” the paper focuses solely on video retrieval. There is no exploration or evaluation of the method on video summarization or question answering tasks.

**Questions:**

1. The list of related works in the paper seems somewhat outdated. Could you provide a more comprehensive review of recent studies, particularly those related to token sampling and pruning methods that enhance efficiency for dense spatio-temporal tasks? The potential related work is as follows:
[1] Jin P, Takanobu R, Zhang W, et al. Chat-univi: Unified visual representation empowers large language models with image and video understanding[C]//Proceedings of the IEEE/CVF Conference on Computer Vision and Pattern Recognition. 2024: 13700-13710.
2. Can you address the typos and inconsistencies in the paper, such as the inconsistent references to Figure 1(a) versus “Figure 1a,” to ensure uniformity throughout?
4. Could you clarify a clear breakdown of which components are trained and which are fixed/pretrained? Is it only the neural sampler, or does it also include the positional encoding layer?
5. Can you include the inference time of the ReST system for a fair comparison? In Table 1, only the inference time of Modified TubeDETR is reported, but it would be helpful to include ReST’s inference time as well. Why they are not included if there's a specific reason?
6. Could you provide more quantitative results comparing your method with existing approaches?
7. Could you extend your study to include video summarization and question answering tasks to support the broader claims made?

---

### Official Review · Reviewer_vxzu · 2024-11-02

**Soundness:** 3
**Presentation:** 2
**Contribution:** 1
**Rating:** 3
**Confidence:** 5

**Summary:**

The paper introduces LVM-Net, a novel method for efficient long-form video reasoning that uses a fixed-size memory representation to store discriminative patches sampled from input videos, addressing computational resource and GPU memory constraints. LVM-Net employs a neural sampler to identify important memory tokens and requires only a single pass over the video, significantly improving inference efficiency. The method demonstrates an 18x-75x improvement in inference times for long-form video retrieval and question answering on the Rest-ADL dataset while maintaining competitive predictive performance.

**Strengths:**

The paper presents a significant strength in its ability to drastically reduce inference times for long-form video reasoning by up to 75x, which is crucial for real-time applications.
Additionally, LVM-Net's efficiency allows it to be deployed on edge devices with limited memory, expanding the potential for practical use cases in video understanding.

**Weaknesses:**

1. The idea presented in the paper could be seen as an innovative trick, but it is not substantial enough to support the entire article. The authors should provide more technical details or analysis to strengthen the contribution.

2. The validation of results on a single dataset seems somewhat limited. You can test on other video reasoning dataset such as MM-Bench, Video-bench and so on.

3. Figure 2 is rough. There are specific elements that need better labeling.

**Questions:**

1. The idea presented in the paper could be seen as an innovative trick, but it is not substantial enough to support the entire article. The authors should provide more technical details or analysis to strengthen the contribution.

2. The validation of results on a single dataset seems somewhat limited. You can test on other video reasoning dataset such as MM-Bench, Video-bench and so on.

3. Figure 2 is rough. There are specific elements that need better labeling.

---

### Official Review · Reviewer_Ekvc · 2024-11-08

**Soundness:** 2
**Presentation:** 3
**Contribution:** 2
**Rating:** 3
**Confidence:** 5

**Summary:**

The paper introduces LVM-Net (Long-Video Memory Network), a new approach for long-form video reasoning, crucial for tasks such as video retrieval and question answering. By using a fixed-size memory representation and a neural sampler for discriminative token selection, LVM-Net improves computational efficiency, enabling an 18x to 75x reduction in inference times compared to existing models while maintaining competitive predictive accuracy.

**Strengths:**

1. The writing is good.
2. The improved efficiency is significant.

**Weaknesses:**

1.	Although the authors improved the efficiency by leveraging stored tokens, the performance of the proposed method is not competitive compared to the existing methods shown in Tab. 2. For example, the proposed LVM-Net only achieves 8.6% compared to 12.8% and 30.0% of TubeDETR and ReST, respectively. The performance of other groups is also not promising. This significantly harms the contribution of this paper.
2.	The authors should include more baselines in the benchmark for a comprehensive comparison, e.g., MeMViT mentioned in the related work. The authors could apply a similar adaptation method done on TubeDETR to cope with long videos. Besides, the authors only compared the proposed neural sampling with random sampling in Tab. 3. However, the authors should benchmark the proposed method with other token efficiency methods mentioned in the related work to highlight the effectiveness of the proposed method.
3.	The novelty of the proposed method seems weak. The proposed fixed-size memory is similar to the replay buffer in the field of continual learning, and the neural sampler (Xie et al., 2019) is borrowed from previous works, rendering the novelty of this paper only left in the proposed auxiliary loss and the architectural design. It would be better for the authors to clarify the novelty of LVM-Net.
4.	In Sec. 4, Neural Sampler. The sentences “The neural sampler outputs scores for k (clip) tokens and m (memory) tokens.” and “The neural sampler outputs m discriminative visual tokens.” are ambiguous since the mentioned output of the same module is different.
5.	It would be better to visualize and analyze the tokens stored in the Memory Bank. It is very important to understand the difference between the stored tokens and Fig. 1-a) since the authors should demonstrate the representativity of the stored tokens.

**Questions:**

1.	The reviewers would like to know if the authors included the running time of inference stage 1 in Tab. 1. If yes, the reviewers would like to know the ratio of running time between inference stages 1 and 2.

---

### Meta-Review · Area_Chair_nfVe · 2024-12-21

**Metareview:**

This paper presents a novel framework for efficient long-form video reasoning. Overall, while the reviewers found the proposed framework interesting, they were concerned that the novelty is weak and that the paper presentation needs to be significantly improved (as several parts are unclear and hard to understand).

There is no rebuttal. The final decision is rejection.

**Additional Comments On Reviewer Discussion:**

No rebuttal is provided and all reviewers have ratings below 5.

---

### Decision · Program_Chairs · 2025-01-22

Reject